# Arctic weather variability and connectivity

Jun Meng[1], Jingfang Fan ®[2,3] ✉, Uma S. Bhatt ®[4,5] & Jürgen Kurths ®[3,4,5,6]

The Arctic's rapid sea ice decline may influence global weather patterns, making the understanding of Arctic weather variability (WV) vital for accurate weather forecasting and analyzing extreme weather events. Quantifying this WV and its impacts under human-induced climate change remains a challenge. Here we develop a complexity-based approach and discover a strong statistical correlation between intraseasonal WV in the Arctic and the Arctic Oscillation. Our findings highlight an increased variability in daily Arctic sea ice, attributed to its decline accelerated by global warming. This weather instability can influence broader regional patterns via atmospheric teleconnections, elevating risks to human activities and weather forecast predictability. Our analyses reveal these teleconnections and a positive feedback loop between Arctic and global weather instabilities, offering insights into how Arctic changes affect global weather. This framework bridges complexity science, Arctic WV, and its widespread implications.

Arctic sea ice is declining and thinning at an accelerating rate due to anthropogenic climate change[1,2]. The overall warming trend is double in the Arctic compared to the global average and is even higher in some Arctic regions[3] due to a phenomenon known as Arctic amplification (AA)[4–8]. Arctic sea ice conditions can affect the Arctic ecosystem, wildlife, hunting, shipping, natural resource exploration and more[9–11]. As one crucial component of the complex Earth system[12,13], changes in Arctic sea ice are found to have statistically observable and dynamic connections with regional and remote climatic phenomena[14–17] (as shown in Fig. 1) through both large-scale atmospheric and oceanic circulations[18–22]. The rapid shrinking of the ice cover has attracted much attention to the Arctic sea ice teleconnections and predictions on seasonal-to-decadal time scales in recent years[23–26]. However, our understanding of the Arctic sea ice variability on weather time scales is still in its infancy[27,28], despite the importance that it has for weather forecasting, the safety of commercial and subsistence maritime activities, the survival of polar mammals and the benefit of polar economics. The impact of day-to-day Arctic sea ice variations has been underestimated in most of the climate models[29]. To resolve this knowledge gap, here we employ complexity-based approaches and the *climate network* framework to investigate the daily weather variability (WV) of Arctic sea ice and assess its connections in relation to climate phenomena on different

spatiotemporal scales. This includes processes such as the Arctic Oscillation (AO), climate change, and weather conditions both locally and at remote locations.

Complexity science employs the mathematical representation used by network science and provides a powerful tool to study the structure, dynamics and function of complex systems[30]. The climate system is a typical complex adaptive system due to its nonlinear interactions and feedback loops between and within different layers and components. In recent years, network science has been applied to the climate system to construct the climate network (CN)[31]. The CN is an innovative tool used to reveal and predict various important climate mechanisms and phenomena[32], including forecasting of the El Niño Southern Oscillation[33,34], the Indian summer monsoon rainfall[35,36], the global pattern of extreme-rainfall[37], the changes in the global-scale tropical atmospheric circulation under global warming[38], teleconnections among tipping elements in the Earth system[39], the Indian Ocean Dipole[40] and other phenomena.

The AO is one of the major modes of atmospheric circulation over the mid-to-high latitudes of the Northern Hemisphere (NH)[41], which influences climate patterns in Eurasia, North America, Eastern Canada, North Africa, and the Middle East, especially during boreal winter[42–44]. The AO index is defined as the leading empirical orthogonal function of NH sea level pressure (SLP) anomalies from latitudes 20°N to 90°N

[1]School of Science, Beijing University of Posts and Telecommunications, 100876 Beijing, China. [2]School of Systems Science/Institute of Nonequilibrium Systems, Beijing Normal University, 100875 Beijing, China. [3]Potsdam Institute for Climate Impact Research, Potsdam 14412, Germany. [4]Geophysical Institute, Department of Atmospheric Sciences, University of Alaska Fairbanks, Fairbanks, AK 99775, USA. [5]College of Natural Sciences and Mathematics, University of Alaska Fairbanks, Fairbanks, AK 99775, USA. [6]Institute of Physics, Humboldt-University, Berlin 10099, Germany. ✉e-mail: jingfang@bnu.edu.cn

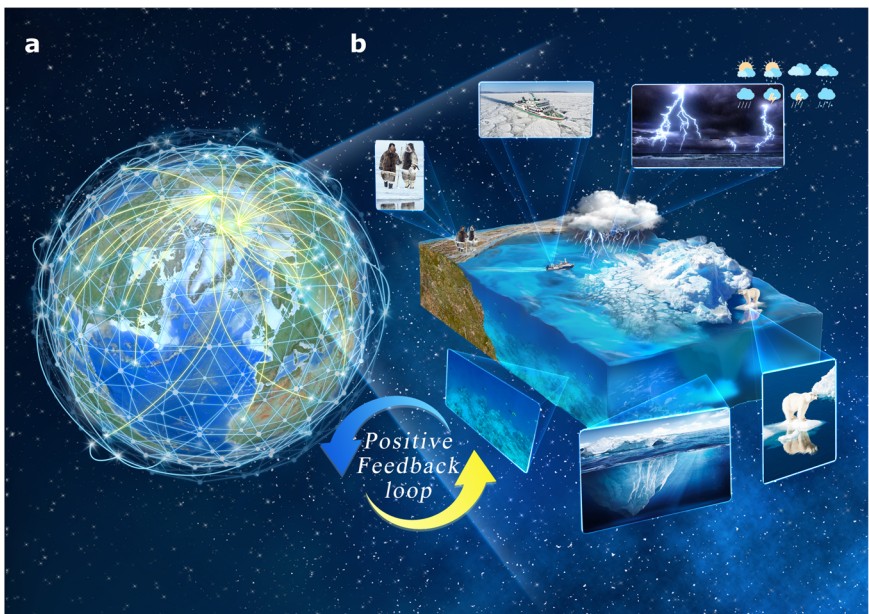

**Fig. 1 | A visual representation of the importance of the Arctic system and its interactions with the global climate system. a** Schematic view of a climate network. Links indicate interactions between different regional climate systems around the globe. The golden links specifically represent teleconnections between the Arctic and regions outside of the Arctic, indicating the influence and connectivity of the Arctic with the rest of the world. **b** illustrates the complex Arctic system, highlighting its various components, such as the cryosphere, biosphere, hydrosphere, and atmosphere. These components interact with each other, and changes in one component can trigger cascading effects and feedbacks in interconnected processes. The circular arrow suggests a positive feedback loop of weather variability (WV) between the Arctic and the rest of the climate system. This indicates that changes in WV in the Arctic can influence weather patterns and climate dynamics in other regions and vice versa. The feedback loop emphasizes the interconnected nature of the Earth's climate system, where changes in one component, such as Arctic WV, can have wide-ranging effects on the entire system.

and is characterized by the shifting of atmospheric pressure between the Arctic and the mid-latitudes.

During the *positive AO phase*, the SLP is below average in the Arctic region and above average in the mid-latitudes. The jet stream is zonal and shifts northward, accompanied by a poleward shift of the storm tracks[45]. In line with these dynamics, our recent findings demonstrate more rapid changes in sea ice and air temperature within the NH's mid-to-high latitudes, as evidenced by a blue-shifted frequency spectrum. This stands in contrast to the relatively stable weather conditions observed in the further southern mid-latitudes, which are characterized by a red-shifted spectrum. These patterns diverge from those seen during the *negative AO phase*, which exhibits higher-than-normal Arctic SLP. To quantify the blue-shift/red-shift effect indicating increased/reduced WV and its geographic distribution, here we introduce two innovative mathematical techniques: the *advanced weighted autocorrelation function method*, i.e., $W_{ACF}$ and the *advanced weighted power spectrum method*, i.e., $W_{PS}$ (see "Methods").

By employing these methods, we are able to demonstrate that the enhanced day-to-day variability in the ice cover across much of the Arctic can be attributed to substantial declines in sea ice area[46]. This has serious consequences, such as the increased risk of severe weather due to climate change[47–50]. Further, it is also possible that changes in the Arctic may have some role in affecting the likelihood of extreme weather conditions globally, although this is subject to ongoing research.

Finally, we statistically verify the existence of these teleconnections between Arctic sea ice and weather conditions in remote global regions via a multivariate climate network framework. These teleconnections can result in a positive WV feedback loop between the Arctic and other global locations. This improves our understanding of the mechanisms that link AA and mid-latitude weather[51].

The presented results and methodology not only encourage a quantitative assessment of the risks posed by extreme weather events but also reveal the existence of pathways that permit the interaction or synchronization among regional and global climate components. Unveiling the interconnections between different climatic elements advances our understanding of climate dynamics and enhances the potential predictability of the Earth's climate system.

## Results

### Linkage of the weather variability and the AO

The WV refers to the irregularity/predictability of the climate data at weather time scales (i.e., hours–days). There are various ways to evaluate data variability/irregularity, such as entropy[52–54], the detrended fluctuation analysis[55,56], correlation dimension[57], Lyapunov exponents analysis[58], etc. However, most of them would be problematic, biased or invalid when dealing with short and noisy data, like weather data. The standard deviation (SD) is an effective metric for quantifying the dispersion of data, but is not a good way to measure irregularity, e.g., the SD of randomly shuffled data is the same as the original. In addition, the autocorrelation function describes the speed at which the self-similarity of a variable decays with time[59], and power spectral analysis[60] allows us to discover periodicity within data. However, a comprehensive evaluation of the autocorrelation and the power spectrum, as well as their dynamic evolution for nonstationary climate data, is still lacking.

Therefore, we present two mathematical methods, $W_{ACF}$ and $W_{PS}$ (refer to the "Methods" section for detailed explanations), to measure the WV within and near the Arctic region on a monthly basis. These methods are applied to the daily data spanning from 1979 to 2019. These functions allow us to quantify the dynamic behavior of the Arctic WV spanning from January 1980 to December 2019. For a given time series, the physical meaning of these metrics is as follows: higher $W_{ACF}$ values indicate a weaker short-term memory; while higher $W_{PS}$ values suggest faster changes (high-frequency variations).

In particular, to better understand their physical interpretations, we generate various nonlinear time series (illustrated in Fig. 2a) using

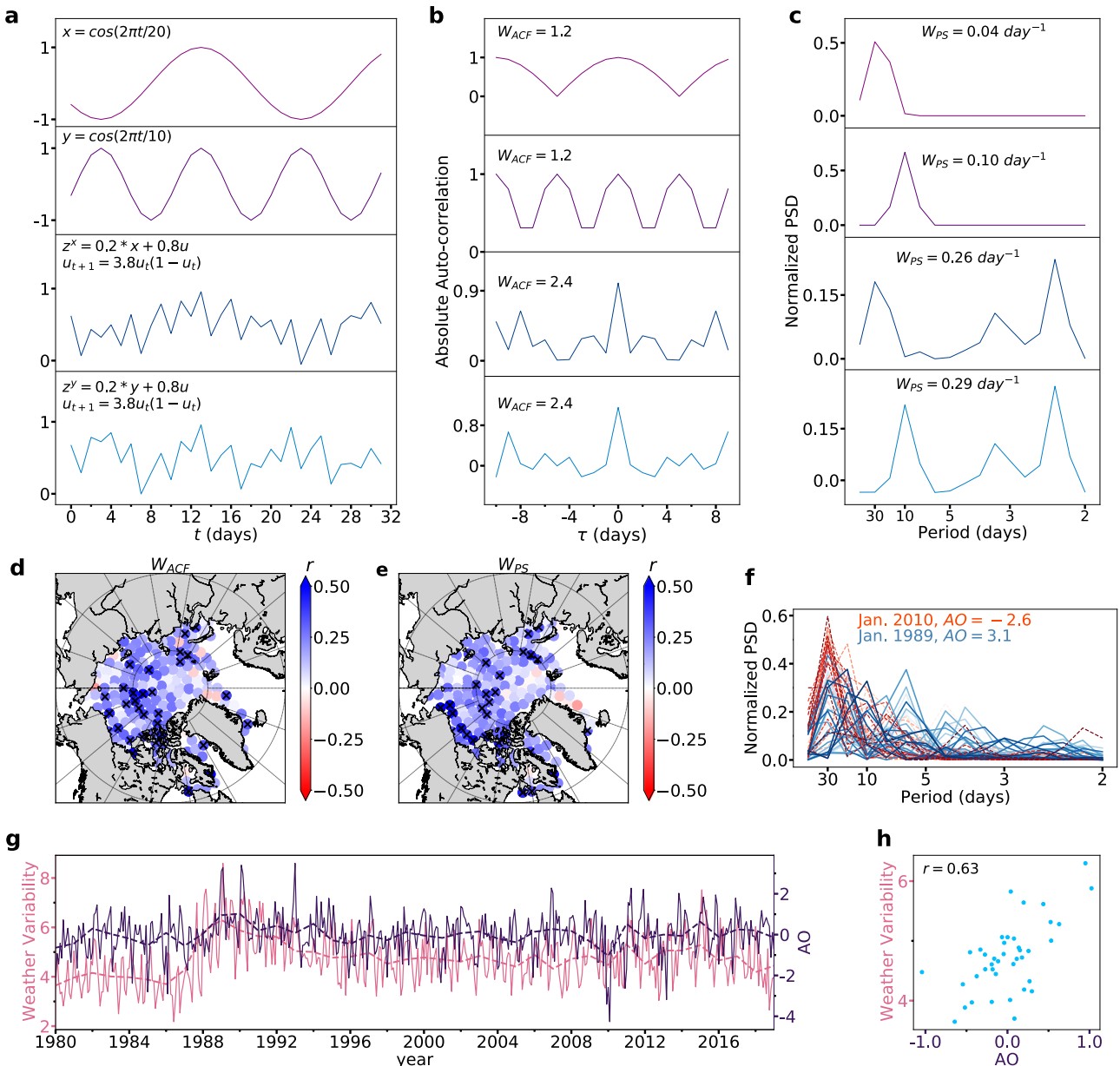

**Fig. 2 | The influence of the Arctic Oscillation on Arctic weather variability.**
**a** shows a set of nonlinear time series generated based on Eqs. (1–4). Auto-correlation functions and weighted autocorrelation function $W_{ACF}$ values are shown in (**b**), while power spectrum density and weighted power spectrum $W_{PS}$ values are shown in (**c**). Correlations between the annual mean of the Arctic Oscillation (AO) index and the $W_{ACF}$ for Arctic sea ice are depicted in (**d**) and for $W_{PS}$ in (**e**). Nodes with correlations significant at the 95% confidence level (Student's $t$-test) are marked with "x". **f** compares the normalized power spectral density (PSD) of sea ice

for nodes marked with "x" in (**e**) during Jan. 1989 (positive AO phase, blue solid lines) and Jan. 2010 (negative AO phase, red dashed lines). **g** The monthly and annual AO index (pink solid and dashed lines, respectively) are plotted against the monthly and annual $W_{ACF}$ index (dark blue solid and dashed lines, respectively) averaged over the nodes marked with "x" in (**d**). **h** The relationship between the AO and $W_{ACF}$ indices is further illustrated through scatter plots of annual indices. The correlation coefficient ($r$) between these two indices is 0.63, with a $p$-value of $5.5 \times 10^{-6}$. Source data are provided as a Source Data file.

the following dynamical equations,

$$x_t = \cos(2\pi t/20), \tag{1}$$

$$y_t = \cos(2\pi t/10), \tag{2}$$

$$z_t^x = 0.2x_t + 0.8u_t, \tag{3}$$

$$z_t^y = 0.2y_t + 0.8u_t, \tag{4}$$

where $t$ is measured in days and belongs to the interval [1000, 10000]. The function $u_{t+1} = \mu u_t(1 - u_t)$ represents a nonlinear logistic function, where we have set the parameter $\mu = 3.8$ and $u_0 = 0.01$. This specific parameter configuration results in the generation of chaotic behavior[61]. Mathematically, Eqs. (1) and (2) are two periodic functions but with periods of 20 days and 10 days, respectively. On the other hand, Eqs. (3) and (4) consist of a combination of a periodic term and a chaotic term (as shown in Fig. 2a).

Therefore, strictly speaking, the value of $W_{ACF}$ for $z_t^x$ ($z_t^y$) is higher than $x_t$ ($y_t$), indicating weaker short-term memory caused by the presence of the chaotic term $u_t$. The value of $W_{PS}$ for $y_t$ ($z_t^y$) is higher than $x_t$ ($y_t$), indicating faster changes attributed to the distinct periods of the

periodic term. We analyzed 31 consecutive data points from each sample, representing one month of climate data, using $W_{ACF}$ and $W_{PS}$ methods. The results presented in Fig. 2b, c support our theory, demonstrating that $W_{ACF}$ and $W_{PS}$ effectively describe the variability (both *disorder* and *frequency*) of the given time series.

In this study, we assess the WV of Arctic sea ice by employing $W_{ACF}$ and $W_{PS}$ based on the sea ice extent dataset (daily data from 1979 to 2019, see "Data" section for details). The results of our analysis are presented in Fig. 2d–h. The positive correlation coefficient $r$, represented by blue shading in Fig. 2d or e, suggests a positive correlation between the annual mean of the $W_{ACF}$ or $W_{PS}$ values with the AO index. We observe that both $W_{ACF}$ and $W_{PS}$ exhibit higher values, implying faster and more irregular day-to-day changes in ice cover during the *AO positive phases* compared to *AO negative phases*. This is particularly true in certain Arctic regions such as the Canadian Archipelago, Beaufort Sea, and Central Arctic. To illustrate the effect of the AO on the $W_{PS}$, we show that the power spectrum for Arctic sea ice during the positive AO phase, e.g., Jan. 1989, is significantly blue-shifted compared to the negative AO phase, e.g., Jan. 2010 (see Fig. 2f). Similarly, to illustrate the effect of the AO on the $W_{ACF}$, we show that the time series for the AO index and $W_{ACF}$ are significantly synchronized during the period 1980-2019 (as shown in Fig. 2g, h). Moreover, we reveal that the climatic effects of the AO are more prominent in the winter-spring seasons than in summer-autumn (see Supplementary Figs. S1 and S2).

The underlying physical mechanism can be attributed to the characteristic atmospheric behavior of the AO and the close interactions between the Arctic sea ice and the surface atmosphere. During the positive phases of the AO, the northward shift of the jet stream and the displacement of storm tracks to higher latitudes[62] (as illustrated in Supplementary Fig. S3) contribute to increased regional weather variability in the mid-to-high latitudes of the Northern Hemisphere (NH). Consequently, this leads to higher values of $W_{ACF}$ and $W_{PS}$ in the air temperature data, as depicted in Fig. 3 and Supplementary Fig. S3. Conversely, during negative AO phases, the Arctic region exhibits lower values of $W_{ACF}$ and $W_{PS}$ in air temperature, indicating higher predictability of weather[63,64]. Simultaneously, in the mid-latitudes of the NH, the $W_{ACF}$ and $W_{PS}$ of air temperature increase with the occurrence of significant weather events such as cold events, frozen precipitation, and blocking days[62]. This occurs as the zonal wind weakens during negative AO phases, as depicted in Fig. 3 and Supplementary Fig. S3.

In particular, as shown in Supplementary Fig. S4, we observe notable connections between the AO and WV within specific regions of the Southern Hemisphere (SH). These correlations between WV and the AO in the SH are noteworthy due to their significant seasonal fluctuations. A more detailed examination reveals a correspondence between these correlations and the position of the jet stream. Areas along the jet stream belt in the SH demonstrate significant correlation coefficients, and these coefficients display a marked seasonal divergence, especially between summer and winter. This pattern hints at a mutual influence between the positions and strengths of the jet streams in the northern and southern hemispheres, thereby underscoring the interconnectedness of atmospheric dynamics across hemispheres and the global implications of the AO. Moreover, the seasonality of these correlations emphasizes the importance of considering seasonal variations and their interaction with the AO when assessing the impacts of this atmospheric phenomenon on weather patterns.

Overall, the analysis of $W_{ACF}$ and $W_{PS}$ provides an additional quantitative approach to understanding the response of Arctic sea ice and the atmosphere to the AO. This methodology can be utilized to assess the risk of extreme events in the mid-to-high latitudes of the NH, enhancing our understanding of the relationships between the AO, weather variability, and the potential occurrence of extreme events in these regions.

## Increased irregularity of Arctic sea ice cover

Using the past four decades (1980–2019) of data, our analysis reveals an evident upward trend in the weather variability linked to the day-to-day changes in the ice cover within a substantial region of the Arctic, with a specific concentration in and near the Central Arctic (see Fig. 4b, c). This increasing trend is supported by a notable rise observed in both the $W_{ACF}$ and $W_{PS}$ values. Figure 4 demonstrates that during this period, a significant proportion of Arctic nodes show a significant increase in the values of $W_{ACF}$ or $W_{PS}$ when comparing values for the same months in earlier years (see Fig. 4a).

The observed increasing trend in WV can be attributed to two main factors. First, advancements in remote sensing and data analysis technology have led to improved data resolution and accuracy throughout the observation record. Second, the rise in air temperature has played a substantial role in the thinning of sea ice[65], with the rapid decline in multiyear ice cover being particularly influential in enhancing WV levels. The multiyear sea ice has been defined as the ice that survives at least one summer melt and represents the thick sea ice cover, while the first-year ice refers to the ice that has only one year of growth or less. As more perennial ice cover is replaced by younger and thinner ice cover, the regional ice cover becomes more fragile and vulnerable to fluctuations in air temperature or other forces[46].

Supporting our findings, Supplementary Fig. S5 illustrates a substantial decline in sea ice thickness within the same period (1980–2019) in regions exhibiting a significant enhancing trend of WV. To further examine the relationship between increased WV and decreased multiyear sea ice cover in the Arctic, we conducted additional analysis (Supplementary Fig. S6). These results show that sub-regions of the Arctic with no significant trends in ice cover thinning also exhibit no significant trends in increasing weather variability. However, in most regions experiencing significant trends of ice cover thinning, there is a notable increase in weather variability. These findings provide further evidence for the connection between declining thick perennial ice cover and heightened WV in the Arctic region.

Moreover, we observe a higher occurrence of significant trends indicating enhanced instability during the melt season under global warming, as shown in Fig. 4a. This is because during the melt season (Apr.-Aug.), sea ice declines and fluctuates more dramatically than in other seasons. The distinct variability in weather patterns across different calendar months suggests the presence of a predictability barrier, particularly in June when both $W_{Auto}$ and $W_{PS}$ reach their peak values. The enhanced weather variability during this period makes weather prediction more challenging and less predictable. Additionally, an intensification of summer Arctic storm activity is also anticipated as the land-sea thermal contrast increases under global warming[66–68]. This could increase the WV in both the ocean and the atmosphere.

## Arctic-global teleconnection patterns

Next, we propose the *multivariate climate network* approach to statistically reveal the potential teleconnection patterns between Arctic sea ice (Supplementary Fig. S7a) and the global air temperature field (Supplementary Fig. S7b); see more details in "Methods". Unlike traditional climate network approaches that focus on single climate variables (see refs. 32,69 and references therein), our approach constructs climate networks with links connecting nodes in the Arctic (Supplementary Fig. S7a) to nodes located elsewhere in the world (Supplementary Fig. S7b). Each link in the network quantifies the similarity of the temporal evolution between two different climate variables: Arctic sea ice and global air temperature. The statistical significance of each link is determined through a comparison with the null model (refer to "Methods" for specifics).

By applying the multivariate climate network, we are able to detect significant synchronizations between Arctic sea ice and global air temperature variations. In Fig. 5a, a typical link is observed,

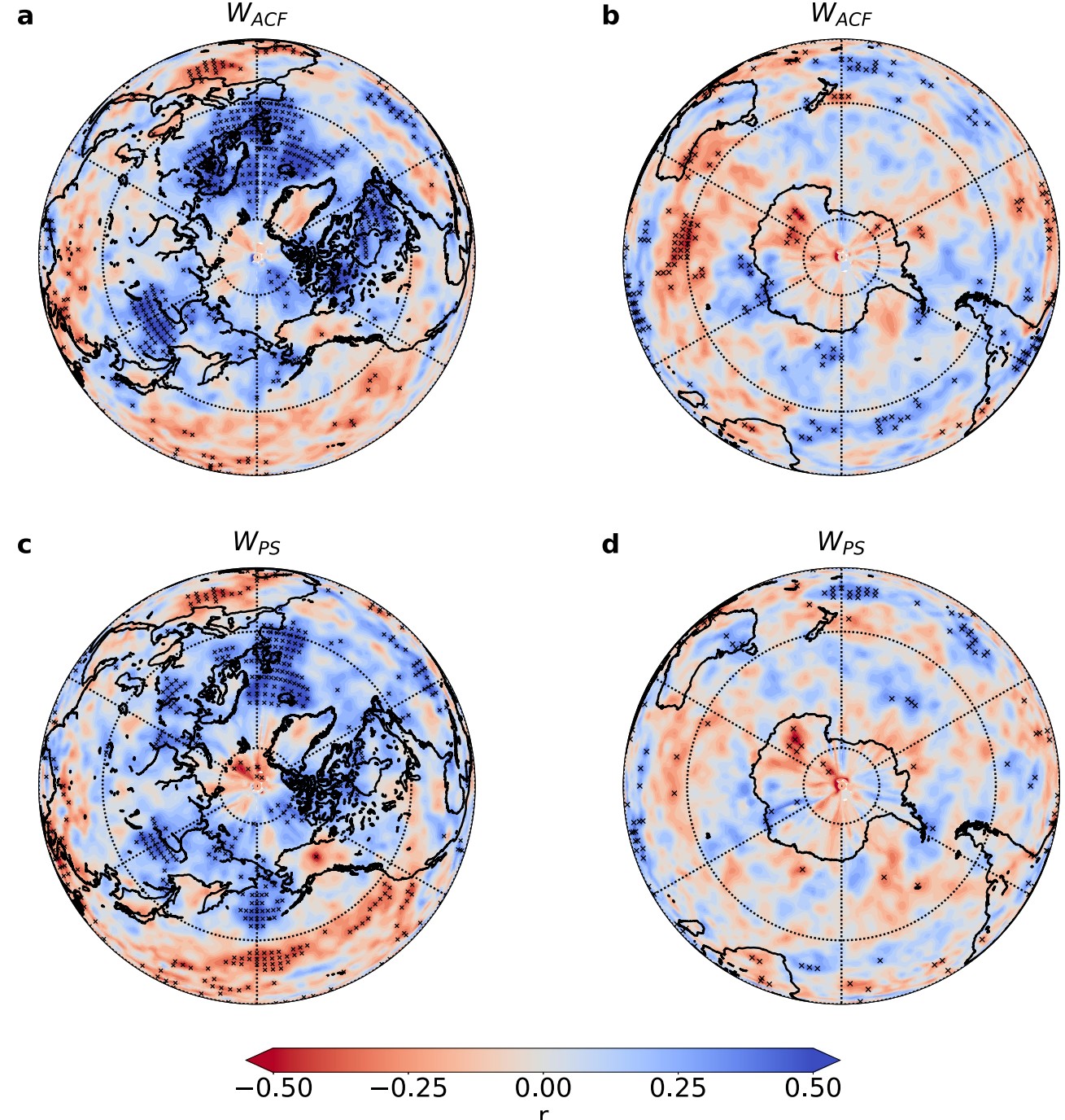

**Fig. 3 | Relationships between the Arctic Oscillation (AO) and weather variability. a** and **b** show correlation maps between the annual mean of the AO index and weighted autocorrelation function $W_{ACF}$ values for air temperature at the 850 hPa pressure level from 1980 to 2019. Similarly, **c** and **d** display the correlation maps for the weighted power spectrum $W_{PS}$. The "x" in each panel indicates regions with correlations significant at the 95% confidence level (Student's $t$-test). The color bar represents the cross-correlation coefficients, denoted as $r$. Source data are provided as a Source Data file.

indicating a strong synchronization between the daily sea ice cover for one Arctic node and the air temperature for another remote global node during December 2018 (the corresponding time series are depicted in Supplementary Fig. S8). Specifically, Supplementary Fig. S8 illustrates that temperature changes at the node located in the Sichuan Province of Southwest China (30°N, 105°E) precede variations in sea ice at the Arctic node (77.5°N, 160°E) by a span of five days. This indicates that the evolution of air temperature in Southwest China can influence Arctic sea ice anomalies. To gain deeper insights into the relationship between sea ice and distant air temperature variability, we

utilize the *shortest path* method (see "Methods" for more details) to identify the most likely teleconnection propagation path.

We discover a potential propagation path for this teleconnection, depicted in orange in Fig. 5a, which corresponds to negative wind anomalies from Southwest China to the Arctic. Additionally, we consider the feedback in the opposite direction. However, we observe a relatively weaker connection, specifically following a straight line from the Arctic to Southwest China via Eastern Russia and Mongolia. From a meteorological perspective, our analysis is highly consistent with wind climatology (refer to the background information in Fig. 5a). These two

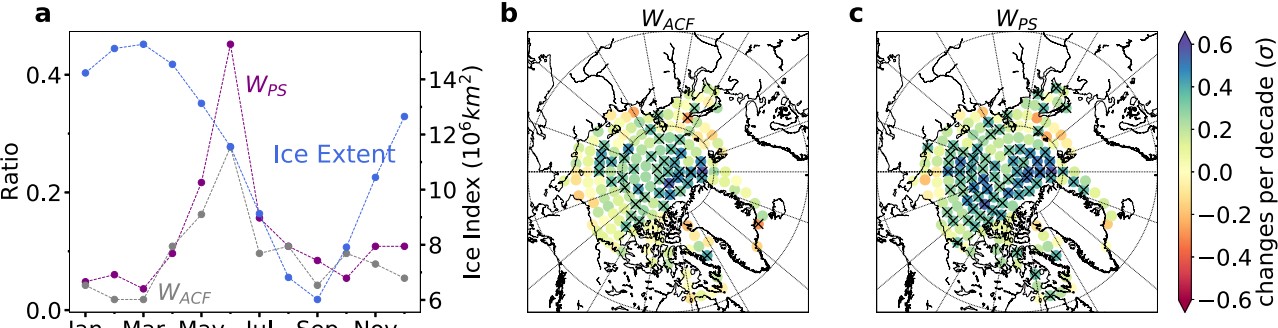

**Fig. 4 | Weather variability of Arctic daily sea ice cover in June. a** shows the ratio of nodes with statistically significant increasing trends in weighted autocorrelation function $W_{ACF}$ (gray) and weighted power spectrum $W_{PS}$ (purple) for each month, overlaid with areas of at least 15% ice cover (blue) from 1980 to 2019. The weather variability is quantified on a monthly basis using the daily data from 1979 to 2019. **b** and **c** display changes per decade as multiples of one standard deviation ($\sigma$) for $W_{ACF}$ and $W_{PS}$ of Arctic nodes in June. The "x" in panels (**b**) and (**c**) represent regions with significant trends at the 95% confidence level (Student's $t$-test). Source data are provided as a Source Data file.

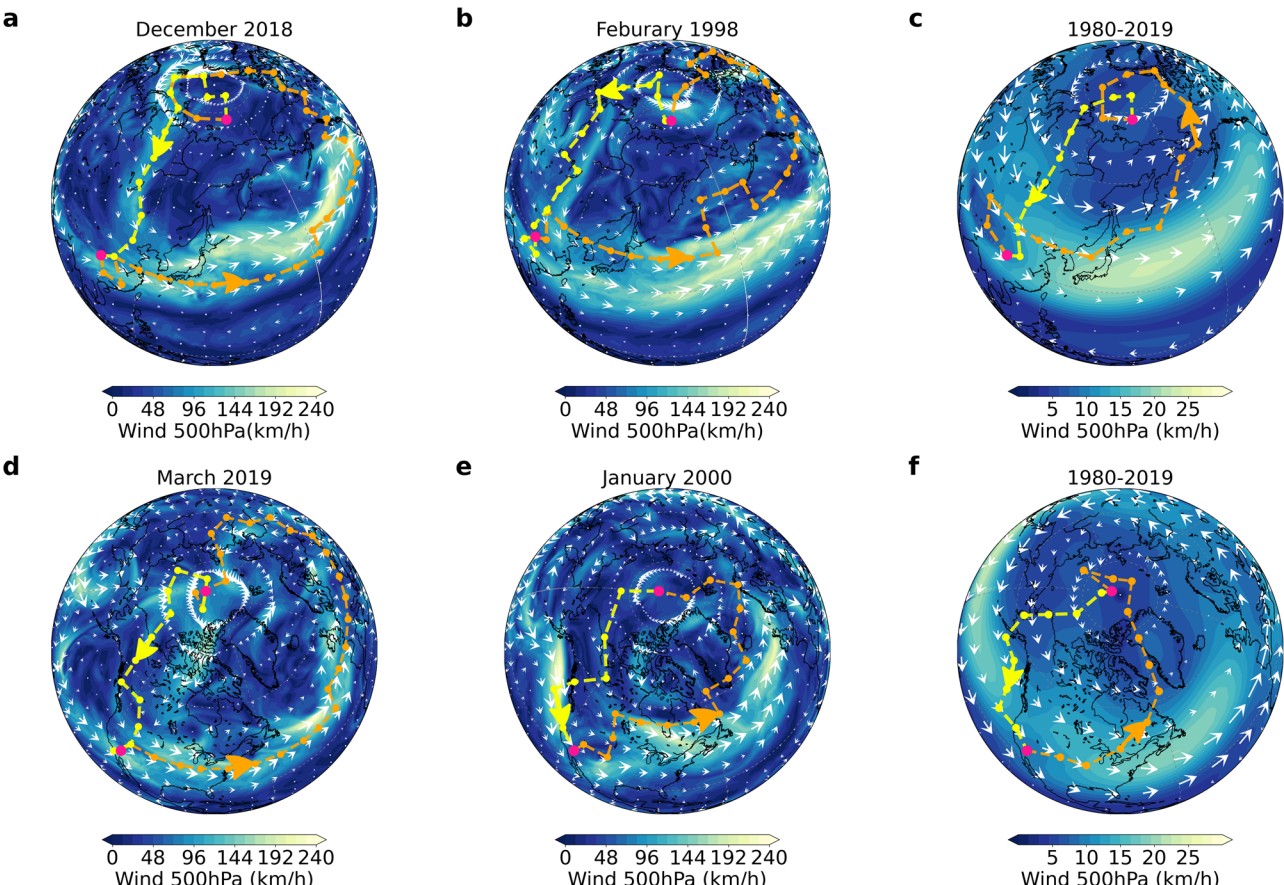

**Fig. 5 | Visualization of the propagation pathway of teleconnection in the climate network. a** illustrates the teleconnection pathway (dashed curves with arrows) between an Arctic node (77.5°N, 160°E) and a global node at (30°N, 105°E) in Southwest China, as observed in December 2018. **b** depicts the same pathway but for the month of February 1998. **c** presents the same teleconnection, but spans a longer period, covering the recent 40 years (1980–2019). The colors and white arrows represent the magnitudes and directions of the 500 hPa winds on a specific day within the network period in (**a**) and (**b**), while in (**c**), they represent the temporal average over the recent 40 years. Panels **d**–**f** showcase analogous information as (**a**–**c**), but for another teleconnection link between California, United States (35°N, 115°W) and the Arctic (87.5°N, 165°W). Source data are provided as a Source Data file.

propagation paths form an interactive loop that suggests a large-scale atmospheric WV feedback between the Arctic and Southwest China.

Remarkably, we have identified similar propagation paths in other months, such as February 1998 (Fig. 5b and Supplementary Fig. S9). To further substantiate the robustness of this feedback loop, we construct climate networks based on the similarity between Arctic and global air temperature variations during longer time durations. Specifically, we analyze time series spanning the past 40 years (1980–2019) (as shown in Fig. 5c), as well as time series spanning the most recent 30, 20, or 10 years (as illustrated in Supplementary Fig. S10a–c). Strikingly, these extended time series analyses consistently reveal that the propagation paths exhibit similar patterns akin to those observed during individual months. In particular, we observe a negative anomalous wind flow from Southwest China to the Arctic, followed by a nearly straight-line

path from the Arctic back to Southwest China, thereby forming a feedback loop. This consistency across different time spans underscores the robust nature of these teleconnections between the Arctic and Southwest China, as they closely follow large-scale atmospheric circulation patterns.

Moreover, we observe another notable teleconnection (California and Arctic) as depicted in Fig. 5d–f. This teleconnection pathway, similar to the previous one observed between Southwest China and the Arctic, is characterized by negative zonal wind anomalies, indicating that fluctuations in air temperature in California can influence Arctic sea ice. Conversely, changes in Arctic sea ice can also impact temperature fluctuations in California, albeit to a lesser extent, along upper wind routes in the opposite direction (more details are shown in Supplementary Fig. S11). The teleconnection path between the Arctic and California, as depicted in Fig. 5d for March 2019, demonstrates similar patterns in other months, such as January 2000 (see Fig. 5e and Supplementary Fig. S12). These patterns remain consistent throughout the time spans of the past 40 years (see Fig. 5f), as well as 30, 20, and 10 years (Supplementary Fig. S10d–f). These teleconnections persist across different time periods and reinforce the potential for Arctic sea ice decline to contribute to droughts and wildfires in California[70].

The synchronization of day-to-day weather between the Arctic and other regions can favor positive feedbacks in WV, whereby an increase in WV/instability of Arctic sea ice may amplify the risk of extreme weather conditions in distant global regions. Conversely, impacts from global regions can also induce unstable weather conditions in the Arctic. These mutual interactions emphasize the potential for cascading effects within the Earth's climate system, where changes in one region can amplify weather variability and contribute to consequential weather events in remote areas.

## Discussion

In summary, we have introduced mathematical methods, namely $W_{ACF}$ and $W_{PS}$, as tools to quantify the changes of WV in climate data. These methods allow us to analyze the temporal variations and irregularities in WV, providing insights into the short-term dynamics of this important climate variable. By applying these methods, we have identified the significant influence of the AO on day-to-day changes in Arctic sea ice and WV in mid-to-high latitude regions of the NH. This influence is attributed to shifts in the location of the jet stream and storm-steering associated with different phases of the AO. Furthermore, our analysis reveals that over the past 40 years, the variability of Arctic sea ice on weather time scales has increased, partially attributable to the melting of thick perennial sea ice. This finding underscores both the profound impact of climate change and the role of internal climate variability on the Arctic region[71,72].

To analyze the teleconnections of Arctic weather, we have constructed multivariable climate networks that connect Arctic sea ice with the global air temperature field. By employing the shortest path method, we have identified teleconnection paths and positive feedback loops of WV. In particular, a more meandering jet stream has the potential to enhance the interchange of air masses between higher and lower latitudes, thereby establishing a connection between the Arctic and lower latitudes[73–77]. The reduced stability of Arctic sea ice can lead to unstable weather conditions and reduce the accuracy of weather forecasts[78] globally via Arctic-global teleconnection feedback loops.

Positive feedback loops are central to our understanding of climate systems as they can either amplify or reduce the impact of a particular climatic phenomenon, leading to potentially larger changes in the global climate. Our findings provide valuable insights into the physical mechanisms linking the Arctic Amplification (AA) and the global climate system. They also highlight the significant global impacts of Arctic WV on human and natural systems, particularly under climate change conditions[6,51].

The Arctic region plays a crucial role as a barometer of global climate change, and the ongoing loss of Arctic sea ice is approaching a tipping point with far-reaching implications for Earth's climate[79]. In addition to its immediate utility in analyzing WV dynamics and assessing global impacts, our framework can be applied to studying the synchronicity of connectivity among remote global regions, forecasting sea ice conditions, and assessing systemic risks associated with complex subsystem interdependencies. This has important implications for systemic risk-informed global governance.

Overall, our research enriches the understanding of the dynamic interactions between the Arctic and the global climate system and highlights the need for comprehensive strategies to address the impacts of Arctic changes on a global scale.

## Methods

### Data

This study primarily utilizes two variables: the daily sea ice cover and the air temperature at the 850 hPa pressure level at 0 hr (UTC). The sea ice cover refers to the fraction of a grid box that is covered by sea ice. It provides information on the extent of sea ice within each grid box. The air temperature at the 850 hPa pressure level represents the atmospheric temperature at a specific vertical level above the Earth's surface. This pressure level is chosen because it is just above the boundary layer, which helps avoid direct interactions between the sea ice and the surface atmosphere[26]. These data were extracted from the ERA5 reanalysis datasets, which can be obtained from the European Centre for Medium-Range Weather Forecasts (ECMWF) website[80]. The spatial resolution of the data is 2.5° in both the zonal (latitude) and meridional (longitude) directions.

To ensure global coverage, we selected a total of 8040 grids from the air temperature datasets, distributing them approximately equally across the globe (refer to Supplementary Fig. S7b for further details). Within these selected grids, we identified 377 grids located in the Arctic region over the ocean, which exhibited nonzero sea ice cover for at least one day (refer to Supplementary Fig. S7a for visualization). For each calendar year $y$ and for each grid point, we calculated the anomalous value for each calendar day $t$ by subtracting the original value from the corresponding climatological average and then dividing by the climatological standard deviation. The calculations for the climatological average and standard deviation were based on data spanning the period from 1979 to 2019. Leap days were excluded to maintain consistency in the calendar year duration and simplify the analysis.

The wind data at 500 hPa pressure level was extracted from ERA5 reanalysis datasets, with a temporal resolution of daily and monthly.

The AO index data was obtained from the following URL: https://www.cpc.ncep.noaa.gov/products/precip/CWlink/daily_ao_index/monthly.ao.index.b50.current.ascii [Accessed in Sep. 2021].

The Arctic Sea Ice extent, sourced from the National Snow and Ice Data Center (NSIDC) and depicted as the blue curve in Fig. 4a, was acquired from the URL: https://nsidc.org/data/g02135/versions/3 [Accessed in Jan. 2021]. The Sea Ice extent data product is based on gridded fields of sea ice concentration data derived from passive microwave radiometers and is commonly used for monitoring and analyzing changes in Arctic sea ice extent over time[81].

The dataset used to analyze the thinning of ice cover in the Arctic is sourced from NSIDC and obtained from the URL: https://nsidc.org/data/nsidc-0611/versions/4. It provides weekly estimates of sea ice age for the Arctic Ocean, which are derived from remotely sensed sea ice motion and sea ice extent. The dataset covers the time period from January 1984 to December 2021, with a spatial resolution of 12.5 km by 12.5 km. In our analysis, the ratio of thin ice cover refers to the fraction of the area within a specific Arctic sub-region (as illustrated in Supplementary Fig. S6) that is covered by first-year ice or has a sea ice

concentration of less than 15%. A rising proportion of thin ice signifies the melting of the Arctic's ice cover under global warming.

## Assessing weather variability

The autocorrelation function (ACF) is widely used to measure the memory of a time series and reveals how the correlation between any two values of the signal changes as their time lag[59]. For a given time series denoted as $x_t$, the ACF is defined as follows:

$$C(\tau) = \frac{\text{Cov}(x_t, x_{t+\tau})}{\sqrt{\text{Var}(x_t)\text{Var}(x_{t+\tau})}}, \qquad (5)$$

where $\text{Cov}(\mathbf{X}, \mathbf{Y}) = \text{E}[(\mathbf{X} - \text{E}[\mathbf{X}])(\mathbf{Y} - \text{E}[\mathbf{Y}])]$ and $\text{Var}(\mathbf{X}) = \text{E}[\mathbf{X^2}] - \text{E}[\mathbf{X}]^2$. If $x_t$ is completely uncorrelated, for example, a white noise process, $C(\tau)$ is zero at all lags except a value of unity at lag zero ($\tau = 0$). A correlated process, on the other hand, has nonzero values at lags other than zero, which indicates a correlation between different lagged observations. In particular, the short-range memory of $x_t$ is described by $C(\tau)$, which declines exponentially

$$C(\tau) \sim \exp(-\tau/\tau^*), \qquad (6)$$

with a characteristic time scale, $\tau^*$. For long-range memory, $C(\tau)$ declines as a power-law

$$C(\tau) \propto \tau^{-\gamma}, \qquad (7)$$

with an exponent $0 < \gamma < 1$. However, a direct calculation of $C(\tau)$, $\tau^*$ and $\gamma$ is usually not appropriate due to noise superimposed on the collected data $x_t$ and due to underlying trends of unknown origin[82]. To overcome these challenges, we propose an advanced weighted autocorrelation function, $W_{ACF}$, to quantify the memory strength, encompassing both short and long-range correlations in the time series. It is defined as follows:

$$W_{ACF} = \frac{\max(|C(\tau)|) - \text{mean}(|C(\tau)|)}{\sqrt{\text{Var}(|C(\tau)|)}} \equiv \frac{1 - \text{mean}(|C(\tau)|)}{\sqrt{\text{Var}(|C(\tau)|)}}. \qquad (8)$$

Here, $\max(|C(\tau)|)$ and $\text{mean}(|C(\tau)|)$ represent the maximum and mean values of the absolute autocorrelation function $|C(\tau)|$, respectively. $\tau$ belongs to the interval $[-\tau_{max}, \tau_{max}]$, which represents the range of time lags considered. In the present work, we consider a maximum time lag of $\tau_{max} = 10$ days since we are considering the day-to-day changes in data at the time scale of weather forecasting, i.e., within two weeks. Equation (8) describes the fluctuations of the ACF, and its values reveal the strength of memory, i.e., a higher (smaller) $W_{ACF}$ indicates a weaker (stronger) correlation and results in a weak (strong) memory. For example, white noise has a maximum value $W_{ACF} = (2\tau_{max} + 1)\sqrt{\frac{2\tau_{max}}{2\tau_{max}+1}}$. Other examples are described in Fig. 2.

An essential advantage of our method is that it captures the memory strength of the data and provides a robust measure of irregularity, even with limited data points. In contrast, many other methods, such as entropy-based measures and detrended fluctuation analysis (DFA), suffer from reduced accuracy or validity when applied to shorter time series. This aspect makes our method particularly advantageous in situations where data availability is limited, enabling meaningful analysis and assessment of irregularity despite the shorter time series. This also enables the examination of how irregularity evolves and changes over time, uncovering potential trends or patterns in the data. Another significant advancement of our method is its ability to address problematic nonstationarities.

To demonstrate the effectiveness of the $W_{ACF}$ method in eliminating the influence of data length and nonstationarity, we apply our method to artificial model data with known properties, the MIX(p) stochastic processes[52] and the Logistic map. The MIX(p) time series, with $p$ ranging between 0 and 1, can be informally described as a sine wave with random noise, where $N \times p$ randomly selected points are replaced with random noise values. This substitution introduces irregularity, which escalates as the $p$ increases. On the other hand, the irregularity of a Logistic map, given by the equation $x_{(t+1)} = \mu x_t(1 - x_t)$, is controlled by the parameter $\mu$. Both the MIX(p) time series and the Logistic maps provide controlled parameters to adjust the level of irregularity, allowing for a systematic evaluation of the performance of the $W_{ACF}$ method across different irregularity levels.

Subsequently, we calculated the $W_{ACF}$ for time series of varying lengths (i.e., $N = 30$ and 100) and different types of trends. Specifically, we generated multiple sets of MIX(p) time series and logistic maps, to which we added strong monotonous trends ($1000t^b$) with varying power $b = 1/2$, 1 and 2, or oscillatory trends ($10\sin(2\pi tf)$) with different frequencies $f = 1/500$ and $1/1000$, following the approach outlined in ref. 82. The results, as depicted in Supplementary Fig. S13, clearly illustrate that the $W_{ACF}$ accurately reflects the irregularity of the data, regardless of strong and slow (monotonic or periodic) trends being superimposed on the raw data. This substantiates the robustness and reliability of our method in managing nonstationarities and capturing the inherent data irregularity.

Additionally, our approach remains effective even for shorter data series with $N = 30$, enabling us to capture and scrutinize patterns of variability effectively. By partitioning the time series into smaller portions, we gain the ability to monitor shifts in data variability over time. This process affords us valuable insight into the dynamic nature of the system under investigation. However, it is important to note that the shortest length that makes the $W_{ACF}$ method valid may vary for different real-world systems, depending on the intrinsic characteristics of the system under study. It is therefore essential to carefully consider the specific properties and dynamics of the data when determining the appropriate segment length for analysis.

The advanced autocorrelation function $W_{ACF}$ sufficiently quantifies the memory for an arbitrary time series but does not reveal any information about the frequency content. For example, Eqs. (1) and (2) are two functions with different periods. Their $W_{ACF}$ values are almost the same, as shown in Fig. 2. To fill this gap, we further develop an advanced power spectrum (PS) method. Based on Welch's method[83], we define the advanced weighted power spectral density $W_{PS}$ as,

$$W_{PS} = \int_f P(f) \times f \, df, \qquad (9)$$

where $P(f)$ is the normalized spectral density and $f$ stands for the corresponding frequency, which can be obtained by Fourier transform. $W_{PS}$ is indeed the weighted mean of $f$ and thus has the same unit as frequency. Notably, a relatively higher value of the $W_{PS}$ indicates a larger ratio of the high-frequency components (i.e., blue shift), see examples shown in Fig. 2.

## Climate networks

Unlike the classical climate network, which only uses single node classification, see refs. 32,84 and references therein, here, we define two types of nodes: global nodes $i$ with the air temperature variable $T_i(t)$ and Arctic nodes $j$ with the Arctic sea ice cover variable $I_j(t)$. We thus have 8040 global nodes (as shown in Supplementary Fig. S7b) and 377 Arctic nodes (as shown in Supplementary Fig. S7a).

We constructed a multivariate climate network for each month from Jan. 1980 to Dec. 2019. To obtain the strength of the links between each pair of nodes $i$ and $j$, we computed, for each month $m$,

the time-delayed, cross-correlation function

$$C_{i,j}^m(\tau) = \frac{\left\langle T_i^m(t)I_j^m(t-\tau)\right\rangle - \left\langle T_i^m(t)\right\rangle\left\langle I_j^m(t-\tau)\right\rangle}{\sqrt{\mathrm{Var}\left(T_i^m(t)\right)\mathrm{Var}\left(I_j^m(t-\tau)\right)}}, \qquad (10)$$

and

$$C_{i,j}^m(-\tau) = \frac{\left\langle T_i^m(t-\tau)I_j^m(t)\right\rangle - \left\langle T_i^m(t-\tau)\right\rangle\left\langle I_j^m(t)\right\rangle}{\sqrt{\mathrm{Var}\left(T_i^m(t-\tau)\right)\mathrm{Var}\left(I_j^m(t)\right)}}, \qquad (11)$$

where the bracket $\langle\,\rangle$ denotes an average over consecutive days during a given month $m$, and $\tau \in [0, \tau_{max}]$ is the time lag. Since we mainly focus on the dynamic WV in the Arctic, here we chose the maximal time lag $\tau_{max} = 20$ days for Eqs. (10) and (11).

We identified the time lag $\theta$ at which the absolute value of the cross-correlation function $|C_{i,j}^m(\tau)|$ reached its maximum. The *weight* of link $(i,j)^m$ was defined as the corresponding value of the cross-correlation function, i.e., $C_{i,j}^m = C_{i,j}^m(\tau = \theta)$. Therefore, the weight of each link could be either positive or negative, but with the maximum absolute value. The sign of $\theta$ indicates the direction of each link; that is, when the time lag is positive ($\theta > 0$), the direction of this link is from $j$ to $i$, and vice versa[85].

Next, we investigated the statistical significance of the link weights in the real networks by comparing them to the shuffled surrogate networks. In the surrogate networks, we calculated the link weights by utilizing two data segments, each consisting of 30 consecutive days. These segments were randomly selected from the period between Jan. 1980 and Dec. 2019. The purpose of this selection was to ensure that there were no actual correlations between the nodes in the temporal dimension.

We constructed 100 surrogate networks using this strategy and established a significance threshold, denoted as $q$, for each link. This threshold for a specific link was determined as the 95th percentile of the absolute weights for the same link in the surrogate networks. For the real network, specifically for a particular month $m$, we classified the link $(i,j)^m$ as significant if its weight exceeded $q$ or fell below $-q$. Mathematically, this can be represented as $|C_{i,j}^m| > q$.

Constructing climate networks spanning longer time spans involves calculating time-delayed cross-correlation functions (Eqs. (10) and (11)) between pairs of nodes using data records ranging from 10 to 40 years. To assess the significance of the links connecting Arctic nodes to global nodes, null model is created by shuffling the order of years in the original climate data. This approach allows for the examination of long-term relationships between climate variables and the identification of robust connections between the Arctic and the globe.

**Teleconnection path mining**

To identify the teleconnection path, we apply the *shortest path* method to complex networks to find the optimal paths in our climate networks. A path is a sequence of nodes in which each node is adjacent to the next one, in particular, in a directed network, the path can follow only one direction. Here, our climate network is based on only one climate variable—air temperature at the 850 hPa pressure level, and we select 726 nodes from the 10,512 nodes[36,39]. This selection is made to manage computational complexity. For each climate network link $(i,j)^m$, we define its cost function value as

$$E_{i,j}^m = \frac{1}{|C_{i,j}^m|}. \qquad (12)$$

The Dijkstra algorithm[86] was used to determine the directed optimal path between a source node $i$ and a sink node $j$ with the following

constraints[39,87]: (1) the distance for each step is limited to less than 1000 km. However, should we fail to identify a path, the distance is then adjusted to 1500 km; (2) the link time delay $\theta \geq 0$, ensuring that all steps have the same direction; (3) the sum cost function value for all collection of links through path $i \longrightarrow j$ is minimal. In this way, we identify the optimal paths that information/energy/matter follow in two-dimensional space.

## Data availability

The ERA5 reanalysis data used here are publicly available at: https://cds.climate.copernicus.eu/cdsapp#!/dataset/reanalysis-era5-single-levels. The Arctic Oscillation data used here are publicly available at: https://www.cpc.ncep.noaa.gov/products/precip/CWlink/daily_ao_index/monthly.ao.index.b50.current.ascii. The Arctic Sea Ice extent data used here are publicly available at: https://nsidc.org/data/g02135/versions/3 and https://nsidc.org/data/nsidc-0611/versions/4. All other data that support the plots within this paper and other findings are provided as a Source Data file and in a Zenodo repository[88].

## Code availability

The analysis codes used in this study have been deposited in Zenodo[88].

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

## Acknowledgements

The authors wish to thank T. Liu for his helpful suggestions. We acknowledge funding support from the Ministry of Science and Technology of China (2019QZKK0906). J.M. and J.F. acknowledge the support of the National Natural Science Foundation of China (Grant No. 12205025, 12275020, 12135003). U.S.B. acknowledges funding support from the National Science Foundation's Office of Polar Programs through Grants OPP-1749081 for the Sea Ice Prediction Network-Phase 2 (SIPN2). J.K. acknowledges funding support from the Sidney Chapman Chair Endowment through the College of Natural Sciences and Mathematics at the University of Alaska Fairbanks.

## Author contributions

J.M., J.F., U.S.B and J.K. designed the research. J.M. performed the analysis, J.M., J.F., U.S.B and J.K. discussed the results and contributed to writing the manuscript. J.F. led the writing of the manuscript.

## Competing interests

The authors declare no competing interests.
