## [Peer Review File · Nature Communications]

Reviewers' comments:

Reviewer #1 (Remarks to the Author):

The paper by Meng et al “Dynamic Arctic weather variability and connectivity” applies several techniques to datasets of Arctic sea ice and Arctic Oscillations to establish patterns and dependencies in the region.

I think the paper has interesting results, but it requires further improvement in a major revision. The English language of the paper need to be improved in many aspects, so I would recommend service of a professional editor.

The title of the paper contains a tautology: variability is already dynamic, so word “dynamic” has to be omitted, and the title should be “Arctic weather variability and connectivity”.

In several places there are contradictions between weather scale (up to 2 weeks) and longer climatic scale (page 6). This should be corrected, as speaking of weather variability is possible only in short-term scale.

The introduced “advanced autocorrelation function method”, according to Equation 8, is not really a function but a single value for each time series. Introducing such a technique in a small supplement section without thorough testing on artificial data is not convincing, in my opinion. The technique requires clear illustration on data with known properties. Claiming that it removes effect of trends (page 16) should be supported by analytical derivation or numerical simulations.

The proposed “advanced power spectrum method” integrates over a range of frequencies. Why then, in Figure 2, panel c, the units in the legend for W_{PS} are 1/day? Was it calculated for a frequency band around 1/day? If so, this should be clearly explained and justified.

It is not clear what the authors mean by ice cover: extent or volume? Please check the dataset documentation. In page 15, the authors say that they use daily sea ice cover of ERA5, but then mention NSIDC sea ice extent – so what sea ice data was actually analysed?

In Figure 4, panel (a) clearly shows monthly data, but the caption discusses daily weather variability – how can this be explained?

In page 8, first line of 2nd paragraph, “its” – whose?

In figure 3, period 1980-2019 – is this period of average or does it mean that two columns correspond to two years, 1980 and 2019?

Reviewer #2 (Remarks to the Author):

Review of Meng et al 2023

“Dynamic Arctic weather variability and connectivity”

This paper uses statistical techniques to examine Arctic weather variability and teleconnections from the Arctic. The statistical approaches are interesting and provide a new perspective on some of these problems but I don't think the current study is coherent and suffers from looking at a few things in little detail. Here are some of the specific conclusions from the paper:

- “Weather variability” is correlated with the Arctic Oscillation: I think this is well supported by the paper.
- Variability is decreasing due to thinner ice cover: The authors demonstrate the decrease in variability but do not show the link between thinner ice cover, they just point to other papers.
- Identify Arctic teleconnections: The authors use an interesting technique here but only apply to it two months from the 40+ year record so it is hard to estimate whether this is robust and representative. I think for the paper to be accepted, I would suggest a much more thorough analysis throughout the entire record.
- Identify feedback loops through Arctic teleconnections: This is the weakest part of the paper. It is hard for me to understand how robust these linkages are, how much of an impact they have and whether this feedback is stabilising or destabilising.

Overall the study presents interesting complexity science approaches as applied to understanding the Arctic but does not present a strong story or advance our understanding significantly. I suggest the authors think about presenting one component of the story in a more robust and detailed way, or submit to a journal which accepts longer papers so that each component of the story can be developed fully.

I note that I do not feel qualified to assess the use of the statistical approaches in this study. I will focus on the climate response which is more in the domain of my expertise.

Line numbers would be appreciated to enable easy reviewing.

P1: “disrupting the balance of nature” is subjective and should be rephrased

P2: Other recent references which may be of interest:

England, M., I. Eisenman, N. Lutsko and T. Wagner (2021), The recent emergence of Arctic Amplification, *Geophysical Research Letters*, 48, e2021GL094086, doi: 10.1029/2021GL094086

Chylek, P., Folland, C., Klett, J. D., Wang, M., Hengartner, N., Lesins, G., & Dubey, M. K. (2022). Annual mean Arctic Amplification 1970–2020: Observed and simulated by CMIP6 climate models. *Geophysical Research Letters*, 49, e2022GL099371. <https://doi.org/10.1029/2022GL099371>

Figure 1: It is hard for the reader to assess the accuracy of the schematic on the left. The graphic is visually appealing but it is hard to evaluate how scientifically relevant it is. The authors repeatedly point to this figure as evidence or an explanation but it is unconvincing.

P5: I am skeptical that the reported Southern Hemisphere impacts of the AO are robust. Can the authors present some mechanism to explain this regional signature?

P7: Choosing two months from the whole record to explore Arctic teleconnections doesn't seem a robust approach.

Reviewers' comments:

Reviewer #1:

The paper by Meng et al “Dynamic Arctic weather variability and connectivity” applies several techniques to datasets of Arctic sea ice and Arctic Oscillations to establish patterns and dependencies in the region.

Response: We appreciate the referee's accurate summary of our study.

I think the paper has interesting results, but it requires further improvement in a major revision.

Response: We are grateful for the referee's positive remarks on our manuscript and his/her constructive suggestion for a major revision. We have worked diligently to incorporate these improvements in our revised manuscript.

The English language of the paper need to be improved in many aspects, so I would recommend service of a professional editor.

Response: We thank the reviewer for their insightful recommendation regarding the language quality of our manuscript. To address this, we have sought the expertise of professional editors to ensure clarity, precision, and readability throughout our manuscript.

The title of the paper contains a tautology: variability is already dynamic, so word “dynamic” has to be omitted, and the title should be “Arctic weather variability and connectivity”.

Response: We thank the referee for pointing out the redundancy in the title. We appreciate this feedback and have revised the title to "Arctic weather variability and connectivity."

In several places there are contradictions between weather scale (up to 2 weeks) and longer climatic scale (page 6). This should be corrected, as speaking of weather variability is possible only in short-term scale.

Response: We are grateful to the referee for identifying this potential source of confusion. Our sincere apologies for any confusion caused by the contradictory references to weather and climate scales in our paper.

To clarify, our investigation of weather variability employs measures of W_{ACF} and W_{PS} on a monthly basis. Each of these metrics represents the irregularity of daily weather fluctuations within a specified month for each grid point, enabling us to track

and analyze the changes in weather variability over time, and interpret the impact of the Arctic Oscillation (AO), as shown in Fig.2g in the main text.

The annual indices were computed by averaging the monthly values of W_{ACF} and W_{PS} over a year (refer Fig. 2 and 3 in the main text). This average provides a consolidated measure of weather variability for the entire year, effectively reducing spurious correlations introduced by seasonality.

Moreover, we explored the correlation between weather variability and AO on a seasonal basis, ensuring we compared months from the same season for accurate analysis (refer Fig. S1-S4 in the supplementary information).

We sincerely hope that this response clarifies our approach and provides a better understanding of the ways we differentiated between weather and climatic scales.

The introduced "advanced autocorrelation function method", according to Equation 8, is not really a function but a single value for each time series. Introducing such a technique in a small supplement section without thorough testing on artificial data is not convincing, in my opinion. The technique requires clear illustration on data with known properties. Claiming that it removes effect of trends (page 16) should be supported by analytical derivation or numerical simulations.

Response: We sincerely appreciate the referee's valuable feedback and acknowledge the importance of validating new methodologies. We understand the concerns raised regarding the "advanced autocorrelation function method" and its validation.

To address this, we have conducted further analysis on artificial model data with known properties to demonstrate the effectiveness of the W_{ACF} method in removing the influence of trends and capturing the inherent irregularity.

Specifically, we applied our technique to two distinct types of time series with known irregularity patterns: the MIX(p) stochastic processes [1] and the Logistic map. The MIX(p) time series, characterized by a length of N and p ranging from 0 to 1, can be informally depicted as a sine wave contaminated with random noise. Here $N \times p$ points selected randomly are replaced with noise values. This substitution introduces irregularity, which escalates as the p increases. Conversely, the irregularity of the Logistic map, described by the equation $x_{t+1} = \mu x_t(1 - x_t)$, is controlled by the parameter μ . Both MIX(p) time series and Logistic map provide controlled parameters to adjust the level of irregularity, allowing for a comprehensive evaluation of the W_{ACF} method across different irregularity levels.

Subsequently, we calculated the W_{ACF} for time series of different lengths ($N = 30$ and 100) and various types of trends. Specifically, we generated multiple sets of MIX(p) time series and Logistic map and imposed strong monotonous trends of

$1000t^b$ with varying powers $b = \frac{1}{2}, 1, 2$, or oscillatory trends of $10\sin 2\pi t f$ with different frequencies $f = \frac{1}{500}, \frac{1}{1000}$, following the approach outlined in Ref. [2]. The results depicted in Fig. 1 in this letter (corresponding to Fig. S13 in the supplementary information) distinctly illustrate the W_{ACF} method's capability to accurately portray data irregularity, regardless of strong and slow (monotonic or periodic) trends being superimposed on the raw data. This substantiates the robustness and reliability of our method in managing non-stationarities and capturing the inherent data irregularity.

Additionally, our approach remains effective even for shorter data series with $N = 30$, enabling us to capture and scrutinize patterns of variability effectively. By partitioning the time series into smaller portions, we gain the ability to monitor shifts in data variability over time. This process affords us a valuable insight into the dynamic nature of the system under investigation. However, it is important to note that the shortest length that makes the W_{ACF} method valid may vary for different real-world systems, depending on the intrinsic characteristics of the system under study. It is therefore essential to carefully consider the specific properties and dynamics of the data when determining the appropriate segment length for analysis.

We have incorporated a detailed description of these analyses and their results in the Method Section (lines 369-395) to provide a comprehensive understanding of the method's validation using artificial data.

We would like to express our sincere gratitude to the referee for the valuable feedback and constructive criticism. Your insights have significantly contributed to the improvement and strengthening of our manuscript.

Fig.1: An intuitive demonstration of the effectiveness of W_{ACF} in quantifying the variability of non-stationary data. The W_{ACF} values are calculated for the MIX(p) model (a,b) and Logistic models (c,d), which exhibit strong monotonous trends of the form $1000t^b$ with varying power values ($b=2, 1, \text{ and } 0.5$), as well as superimposed oscillatory trends of the form $10\sin 2\pi t f$ with different frequencies ($f=1/500, 1/1000$). The W_{ACF} consistently changes with parameters (p and μ) that control the chaotic behavior of the data. N represents the length of the time series, and each W_{ACF} value is obtained by averaging over 100 realizations.

The proposed “advanced power spectrum method” integrates over a range of frequencies. Why then, in Figure 2, panel c, the units in the legend for $W_{\{PS\}}$ are 1/day? Was it calculated for a frequency band around 1/day? If so, this should be clearly explained and justified.

Response: We appreciate the referee for bringing this issue to our attention. The units for W_{PS} in Figure 2, panel c, are indeed 1/day, which represents that it was calculated for a frequency band around 1/day. The choice of this specific frequency band was determined by the nature of our data, which consists of daily measurements. This alignment ensures compatibility and facilitates meaningful comparisons within the same timescale.

To clarify, in our analysis, we employed time series data measured in days, which aligns with the units of the climate data used in our study.

The time series were generated according to the equations provided:

$$x_t = \cos(2\pi t/20) \quad (1)$$

$$y_t = \cos(2\pi t/10) \quad (2)$$

$$z_t^x = 0.2x_t + 0.8u_t \quad (3)$$

$$z_t^y = 0.2y_t + 0.8u_t \quad (4)$$

where t is measured in days and belongs to the interval $[1000, 10000]$. The function $u_{t+1} = \mu u_t(1 - u_t)$ represents the nonlinear logistic function, where we have set the parameter $\mu = 3.8$ and $u_0 = 0.01$. This specific parameter configuration leads to the emergence of chaotic behavior.

It is not clear what the authors mean by ice cover: extent or volume? Please check the dataset documentation.

Response: Thank you for this important clarification. The term "ice cover" used in our study refers to the fraction of a grid box covered by sea ice in the ERA5 dataset. Therefore, it reflects the extent, rather than the volume, of sea ice within each grid box. We have carefully reviewed the dataset documentation and amended the text.

In page 15, the authors say that they use daily sea ice cover of ERA5, but then mention NSIDC sea ice extent – so what sea ice data was actually analysed?

Response: We apologize for any confusion arising from the mention of different datasets in our manuscript. In our analysis, we indeed used both the daily sea ice cover data from ERA5 and the monthly sea ice extent indices from NSIDC. The daily data from ERA5 was used to analyze weather variability, as it provides comprehensive information by combining model data with observations from various sources. This choice ensures that the daily data of sea ice and air temperature are from the same source, enhancing consistency in our analysis, while the monthly data from NSIDC was used to assess the longer-term trend of Arctic sea ice extent. We opted for the NSIDC data for convenience, as it is readily available for download and widely used for monitoring and analyzing changes in Arctic sea ice extent over time.

We have revised the manuscript to clarify the use of each dataset in the revised Data section.

In Figure 4, panel (a) clearly shows monthly data, but the caption discusses daily weather variability – how can this be explained?

Response: We appreciate the referee's attention to detail. In Figure 4, panel (a), the y-axis indeed represents monthly data. However, this monthly data is derived from the analysis of day-to-day weather variability (W_{ACF} and W_{PS}) within each specific month. By comparing the same month, for instance, June, across all years from 1980 to 2019, we can determine the ratio of Arctic nodes where the weather variability exhibits significant increasing (or decreasing) trends over time. In Figure 4(a), the y-axis index of each grey/purple point represents the proportion of nodes with statistically significant increasing trends for the specific month indicated on the x-axis.

In the revised manuscript, we have clarified this point by stating that the weather variability is quantified on a monthly basis using the daily data from 1979 to 2019 (lines 104-106).

In page 8, first line of 2nd paragraph, “its” – whose?

Response: We apologize for any confusion caused by this unclear reference. In the section 'Arctic-global teleconnection patterns,' we have revealed the potential propagation path through which changes in the temperature in Southwest China can affect sea ice variations in the Arctic. Furthermore, we have observed the influence of Arctic sea ice on weather conditions in Southwest China in the opposite direction through large-scale atmospheric circulations. We refer to this phenomenon as a feedback loop.

To provide further clarification, in the revised manuscript, we have improved the sentence as follows: *We discover a potential propagation path for this teleconnection,*

depicted in orange in Fig.5a, which corresponds to negative wind anomalies from Southwest China to the Arctic. Additionally, we consider the feedback in the opposite direction. However, we observe a relatively weaker connection, specifically follows a straight line from the Arctic to Southwest China via Eastern Russia and Mongolia .
(lines 219-223)

In figure 3, period 1980-2019 – is this period of average or does it mean that two columns correspond to two years, 1980 and 2019?

Response: We apologize for any confusion caused. In Figure 3, the period "1980-2019" refers to the time span of the data used for the analysis. It does not mean that each column corresponds to a specific year. The colors in the heatmap indicate the correlation between the annual mean of W_{ACF} or W_{PS} and the Arctic Oscillation index over the entire period from 1980 to 2019. In the revised manuscript, we have clarified this to avoid any misunderstanding.

Reference:

- [1] Pincus, S. Approximate entropy (ApEn) as a complexity measure. *Chaos* 5, 110–117 (1995).
- [2] Kantelhardt, J. W., Koscielny-Bunde, E., Rego, H. H. A., Havlin, S. & Bunde, A. Detecting long-range correlations with detrended fluctuation analysis. *Physica A: Statistical Mechanics and its Applications* 295, 441–454 (2001).

Reviewer #2 :

This paper uses statistical techniques to examine Arctic weather variability and teleconnections from the Arctic. The statistical approaches are interesting and provide a new perspective on some of these problems but I don't think the current study is coherent and suffers from looking at a few things in little detail. Here are some of the specific conclusions from the paper:

Response: We are grateful for the reviewer's positive comment on our statistical methods and their potential to offer fresh perspectives. We acknowledge the concerns raised about the coherence and depth of our study and have made significant efforts to address these points in the revised manuscript.

- *“Weather variability” is correlated with the Arctic Oscillation: I think this is well supported by the paper.*

Response: We appreciate reviewer's positive feedback regarding our analysis.

- *Variability is decreasing due to thinner ice cover: The authors demonstrate the decrease in variability but do not show the link between thinner ice cover, they just point to other papers.*

Response: We appreciate the referee for raising this point. We think the referee meant that: Variability is increasing due to ice cover being thinner. In response to this concern, we have conducted further analysis to substantiate the correlation between the thickness sea ice cover and weather variability (WV). In our study, we used the *sea ice age* (SIA) data to represent the sea ice thickness, higher (lower) SIA means thicker (thinner) ice cover. In Fig. 2 of this letter (which corresponds to Fig. S5 in the supplementary information), we provided a depiction of the considerable decline in SIA by comparing 1985 and 2019 in regions, where we notice a marked increasing trend in WV (denoted by the pink circles in Figs. 2a and 2b).

To further unveil the relationship between increased WV and decreased SIA in the Arctic (12 sub-regions were chosen as shown in Fig. 2c), we conducted an additional analysis, by plotting the temporal evolution of the ratio of area covered by thin ice (i.e. first year ice or ice concentration less than 15%), the averaged W_{ACF} and W_{PS} , as depicted in Fig. 3 (corresponding to Fig. S6 in the supplementary information). In Arctic sub-regions without significant thinning ice cover trends, we also observed no significant trends of increasing weather variability. However, notably, in most regions demonstrating significant thinning ice cover trends, there was a pronounced increase in weather variability.

These findings suggest a potential relationship between changes in sea ice cover and the variability in Arctic weather patterns. We have impended the above aforementioned into the revised manuscript (lines 181-188).

Fig. 2: Decline of the multi-year sea ice cover. **a**, A sample image displaying sea ice age for the week of Jun. 3-10, 1985 (<https://doi.org/10.5067/UTAV7490FEPB>. [Accessed in Sep. 2021]). **b**, The same as **a** for the year 2019. Nodes with a significant trend of enhancing W_{ACF} or W_{PS} as shown in Fig. 4 of the main text, are marked by pink circles. **c**, Depicts the 12 sub-regions of the Arctic. The colored points indicate grid points with non-zero sea ice cover for at least one day during the years from 1979 to 2019, according to ERA5 datasets.

Fig. 3: Temporal evolution of the ratio of area covered by thin ice (i.e., first year ice or ice concentration less than 15%), the averaged W_{ACF} and W_{PS} for 12 subregions shown in Fig. 2c. The dashed lines are the best fitting lines with significant trends.

• *Identify Arctic teleconnections: The authors use an interesting technique here but only apply to it two months from the 40+ year record so it is hard to estimate whether this is robust and representative. I think for the paper to be accepted, I would suggest a much more thorough analysis throughout the entire record.*

Response: We appreciate the referee's insightful feedback. In our study, we have addressed this issue by providing a more detailed analysis of the teleconnections over the entire 40-year record, emphasizing the robustness and representativeness of our findings, covering the period from 1980 to 2019. We apologize if our initial submission did not sufficiently highlight this.

To underscore the robustness of these teleconnections, we have provided a more comprehensive depiction of their persistence across the record. As illustrated in Figs. 4-5, we demonstrate recurring teleconnections between the Arctic and Southwest China, and between the Arctic and California. Additionally, we have examined climate networks spanning the most recent 40, 30, 20, and 10 years. These analyses reveal consistent teleconnection patterns, thereby underscoring their persistent nature.

We appreciate the referee's insightful feedback, as it has helped us to improve the clarity, comprehensiveness, and rigor of our work. We added the above discussions in lines 227-254.

Fig. 4: Visualization of the propagation pathway of teleconnection in the climate network. **a**, illustrates the teleconnection pathway (dashed curves with arrows)

between an Arctic node (77.5°N , 160°E) and a global node at (30°N , 105°E) in Southwest China, as observed in December 2018. **b**, depicts the same pathway but for the month of February 1998. **c**, presents the same teleconnection, but spans a longer period, covering the recent 40 years (1980-2019). The colours and white arrows represent the magnitudes and directions of the 500 hPa winds on a specific day within the network period in **a** and **b**, while in **c**, they represent the temporal average over the recent 40 years. Panels **d-f** showcase analogous information as **a-c**, but for another teleconnection link between California, United States (35°N , 115°W) and the Arctic (87.5°N , 165°W).

Fig. 5: **a-c**, The shortest path method identifies propagation pathways in climate networks constructed from global air temperature data over the most recent 30, 20 and 10 years. These pathways are between a node in Southwest China (30°N , 105°E) and an Arctic node (77.5°N , 160°E). **d-e**, Similar to **a-c**, but for another teleconnection between California (35°N , 115°W) and Arctic (87.5°N , 165°W). The visualization of these pathways features background colors representing the averaged wind speed (500 hPa), with white arrows indicating the prevailing wind direction over the corresponding periods

• Identify feedback loops through Arctic teleconnections: This is the weakest part of the paper. It is hard for me to understand how robust these linkages are, how much of an impact they have and whether this feedback is stabilising or destabilising.

Response: We sincerely appreciate the referee's constructive comments and understand the concerns raised about the robustness, impact, and stability of the

identified feedback loops through Arctic teleconnections. In response, we have conducted further investigations and made significant improvements to better clarify these aspects.

For the robustness of the feedback loops, we chose the teleconnection paths between the Arctic and Southwest China/California as representative examples. We have repeatedly identified similar feedback paths in multiple instances within a one-month period, as shown in Fig. 4a, b, and d, e. These paths closely coincide with wind routes for individual months, as is evident from the background coloration in the figures.

To further demonstrate the robustness of the feedback loops, we expanded our analysis to consider longer time scales. We constructed climate networks based on the similarity between Arctic and global air temperature variations, and examined time series spanning the past 40 years (1980-2019), as presented in Fig. 4c and f. Additionally, we analyzed time series over the most recent 30, 20, and 10 years, as displayed in Fig. 5 (Fig. S10 in the supplementary information).

Remarkably, our analyses over these extended time periods consistently reveal that the propagation paths exhibit similar patterns to those observed during individual months. In particular, we observe a negative anomalous wind flow from Southwest China/California to the Arctic, followed by a nearly straight-line path from the Arctic back to outside, thereby forming a feedback loop.

Additionally, a wavier jet stream may serve as a potential mechanism that can increase the exchange of air masses between high and low latitudes, establishing a linkage between the Arctic and lower latitudes. Moreover, the variations observed in the teleconnection routes at shorter time scales, such as monthly, may be influenced by the inherent chaotic nature of the jet stream on weather time scales.

We have incorporated these improvements and discussions in the revised manuscript (lines 227-254).

Overall the study presents interesting complexity science approaches as applied to understanding the Arctic but does not present a strong story or advance our understanding significantly. I suggest the authors think about presenting one component of the story in a more robust and detailed way, or submit to a journal which accepts longer papers so that each component of the story can be developed fully.

Response: We greatly value the referee's constructive suggestions and recognize the importance of delivering a robust, detailed narrative that significantly advances our understanding of the Arctic.

In light of the referee's comments, we have carried out extensive revisions to bolster the comprehensiveness and depth of our analysis, particularly focusing on the Arctic teleconnections. We have incorporated additional figures and extended our examination to longer time scales, highlighting the persistence and reliability of the identified teleconnection patterns.

We concur with the referee that a more robust and in-depth presentation of our work would enhance its impact. Hence, we have strived to present a clear and coherent narrative that systematically unveils the Arctic complexities, applying complexity science approaches. We thank the referee for his/her valuable guidance throughout the revision process. It has been instrumental in refining our work and enhancing its overall quality and impact. We earnestly hope that the revised manuscript will meet Nature Communications standards and contribute significantly to advancing our collective understanding of the Arctic.

I note that I do not feel qualified to assess the use of the statistical approaches in this study. I will focus on the climate response which is more in the domain of my expertise.

Line numbers would be appreciated to enable easy reviewing.

Response: As the referee's suggestion, we have incorporated line numbers in our revised manuscript. We are thankful for this valuable recommendation.

P1: "disrupting the balance of nature" is subjective and should be rephrased

Response: We appreciate the referee's comment and understand the importance of objectivity in scientific writing.

P2: Other recent references which may be of interest:

England, M., I. Eisenman, N. Lutsko and T. Wagner (2021), The recent emergence of Arctic Amplification, Geophysical Research Letters, 48, e2021GL094086, doi: 10.1029/2021GL094086

Chylek, P., Folland, C., Klett, J. D., Wang, M., Hengartner, N., Lesins, G., & Dubey, M. K. (2022). Annual mean Arctic Amplification 1970–2020: Observed and simulated by CMIP6 climate models. Geophysical Research Letters, 49, e2022GL099371. <https://doi.org/10.1029/2022GL099371>

Response: We greatly appreciate the referee's recommendations and have diligently reviewed the suggested references. We found them to be particularly informative and beneficial to our work. Accordingly, we have integrated these references into our

revised manuscript, enriching our discussion on Arctic Amplification. We thank the referee for their constructive contribution to enhancing our study.

Figure 1: It is hard for the reader to assess the accuracy of the schematic on the left. The graphic is visually appealing but it is hard to evaluate how scientifically relevant it is. The authors repeatedly point to this figure as evidence or an explanation but it is unconvincing.

Response: We greatly appreciate the referee's feedback on Figure 1. We understand the concerns about the accuracy and scientific relevance of the figure. The figure was indeed intended to be a visual representation of the complexity and interconnectivity within the Arctic climate system, rather than a precise diagram of the processes involved.

In response to your comment, we have revised the caption of Figure 1 to better clarify its purpose, emphasizing its role as an illustrative schematic, and not a data-driven diagram. The figure now serves to visually communicate the broad concept of the Arctic system and its teleconnections with the global climate system. We have also modified references to this figure in the manuscript, ensuring we do not cite it as concrete evidence or as a scientific explanation.

We genuinely value your critique and acknowledge that the emphasis of our work should be on the robust scientific analysis and findings presented throughout the manuscript. Although Figure 1 has been retained in the revised version primarily to maintain the overall structure, we are open to the possibility of removing it if you believe it would be more appropriate. We believe that these revisions will alleviate any confusion surrounding the role and interpretation of Figure 1.

Your feedback has been invaluable in refining our work, and we thank you for bringing this matter to our attention.

P5: I am skeptical that the reported Southern Hemisphere impacts of the AO are robust. Can the authors present some mechanism to explain this regional signature?

Response: We appreciate the referee's skepticism and his/her request for a mechanism to explain the reported Southern Hemisphere impacts of the Arctic Oscillation (AO).

As a prominent pattern of climate variability, the AO primarily influences weather patterns in the Northern Hemisphere. The direct impact of the AO on Southern Hemisphere weather and climate patterns is generally not expected to be robust due to the geographical separation. However, the AO might have indirect impacts on the Southern Hemisphere through complex atmospheric and oceanic teleconnections within the global climate system [1]. As the AO modulates the strength and location

of the polar jet stream, it can indirectly influence atmospheric circulation patterns globally. For example, during the positive phase of the AO, the polar vortex is stronger and more stable, which can lead to more zonal (east-to-west) wind patterns and a faster, more confined jet stream. These changes can influence atmospheric wave propagation, potentially altering the transfer of energy and momentum between different latitudes and between the troposphere and stratosphere. In turn, these changes in atmospheric circulation can influence oceanic circulation patterns. For instance, changes in wind patterns can alter surface ocean currents and upwelling patterns, influencing the distribution of heat and nutrients in the oceans. Through these mechanisms, changes in the AO can potentially have effects on climate and weather patterns in the Southern Hemisphere.

To better understand the Southern Hemisphere impacts of the AO in our study, we have included additional analyses, showing the statistical relationship between AO and atmospheric conditions in the Southern Hemisphere (lines 148-159). In Fig. 6 (corresponding to Fig. S4 in the supplementary information), we observe notable connections between the AO and WV within specific regions of the Southern Hemisphere (SH). These correlations between WV and the AO in the SH are noteworthy due to their significant seasonal fluctuations.

A more detailed examination reveals a correspondence between these correlations and the position of the jet stream. In particular, areas along the jet stream belt in the SH demonstrate significant correlation coefficients, and these coefficients display a marked seasonal divergence, especially between summer and winter. This pattern hints at a mutual influence between the positions and strengths of the jet streams in the Northern and Southern hemispheres, thereby underscoring the interconnectedness of atmospheric dynamics across hemispheres and the global implications of the AO.

Considering the seasonality of these correlations is also crucial. The observed changes in correlation coefficients accentuate the need to consider seasonal dynamics and their interaction with the AO when evaluating its influence on weather patterns.

Our findings offer a feasible explanation for the documented impacts of the AO in the Southern Hemisphere. We appreciate the referee's insightful comment, which has spurred a more in-depth exploration and elucidation of this regional characteristic.

Fig. 6: Visualization of the AO's seasonal influences on Southern Hemisphere local weather variability patterns. **a**, Heatmaps depicting correlations between the AO and W_{ACF} , considering only values from the same season Only (i.e., Dec.-Feb for winter, Mar.-May for spring, Jun.-Jul. for summer, and Sep.-Nov. for autumn). **b**, Analogous to **a** but for W_{PS} . **c**, Similar to **a** but for the monthly averaged zonal wind speed at 250 hPa pressure level (U250). **d**, The mean speed of U250 during the period from Jan. 1980 to Dec. 2019. Correlation values with 95% or higher significance are indicated by an "x".

P7: Choosing two months from the whole record to explore Arctic teleconnections doesn't seem a robust approach.

Response: We understand the referee's concern about the robustness of our results based on the examination of a two-month span. In response, we have expanded our analysis to cover the entire period from 1980 to 2019. To demonstrate robustness, we

have pinpointed recurring teleconnections between the Arctic and specific regions across multiple individual months. Further, by comparing these teleconnection pathways with the wind trajectories of the corresponding months, we have observed a remarkable congruity between them.

By incorporating these analyses, we have strengthened the robustness and reliability of our approach in exploring Arctic teleconnections.

We appreciate the referee's critical insight, which has guided us to perform a more comprehensive analysis and enhanced the robustness of our findings.

Reference:

[1] Clara Deser (2000), On the teleconnectivity of the “Arctic Oscillation”, *Geophysical Research Letters*, 27(6), 779-782, doi: 10.1029/1999GL010945

REVIEWERS' COMMENTS

Reviewer #1 (Remarks to the Author):

I am satisfied with the revision and recommend the revised paper for publication.

Reviewer #2 (Remarks to the Author):

The authors have substantially addressed many of my concerns from the original manuscript, however a few issues remain.

L78-79: This sentence about global extreme weather is exaggerated

L107-108: So WACF and WPS are in the same direction (high values mean shorter memory in system?). I don't understand the term 'faster changes'.

L165-167: Point to which figure you are referring to in particular.

FigS6: I find this somewhat difficult to interpret. Could you report the trend numbers somewhere in the figure? ie the slope coefficients.

L249-251: I think the mention of wavier jet stream is not necessary here. This is a hotly debated topic which I personally do not think is very strong, and I do not think the mechanisms suggested by the authors rely on it. Therefore I think including this discussion may weaken the argument.

L254-256: Overstated.

On a side note, I think the authors need to understand that not all changes in the Arctic are driven by climate change, and some can be attributed to internal climate variability. I think the authors should discuss this and what this means for their results.

Dörr, J. S., Bonan, D. B., Årthun, M., Svendsen, L., and Wills, R. C. J.: Forced and internal components of observed Arctic sea-ice changes, *The Cryosphere Discuss.* [preprint],

<https://doi.org/10.5194/tc-2023-29>, in review, 2023.

England, M., A. Jahn, and L. Polvani (2019), Non-uniform contribution of internal variability to recent Arctic sea ice loss, *Journal of Climate*, 32, 4039-4053, doi: 10.1175/JCLI-D-18-0864.1

Reviewers' comments:

Reviewer #1

I am satisfied with the revision and recommend the revised paper for publication.

Response: We thank the referee for his/her fully support publication of our work.

Reviewer #2

The authors have substantially addressed many of my concerns from the original manuscript, however a few issues remain.

Response: We are grateful for the reviewer's constructive feedback and are pleased to hear that our revisions have substantially addressed many of the initial concerns. We have made additional revisions to specifically address the remaining concerns, and these changes are clearly marked and explained in the revised manuscript.

L78-79: This sentence about global extreme weather is exaggerated

Response: We appreciate the reviewer's attention to detail and the concern raised about the original phrasing. In the revised manuscript, we have modified the sentence as, "It is also possible that changes in the Arctic may have some role in affecting the likelihood of extreme weather conditions globally, although this is subject to ongoing research."

L107-108: So WACF and WPS are in the same direction (high values mean shorter memory in system?). I don't understand the term 'faster changes'.

Response: We appreciate the reviewer's insightful question. Indeed, both W_{ACF} and W_{PS} values are in the same direction—higher values indicate higher variability or irregularity within the system. W_{ACF} primarily focuses on the memory of the time series, while W_{PS} is concerned with its frequency content. When we refer to "faster changes," we mean changes occurring at a higher frequency. We have clarified this point in the revised manuscript by explicitly stating that "faster changes" refer to high-frequency variations.

L165-167: Point to which figure you are referring to in particular.

Response: We appreciate the reviewer's suggestion. The text in lines 165-167 is referring to Figure 4 b and c. We have clarified this in the revised manuscript.

FigS6: I find this somewhat difficult to interpret. Could you report the trend numbers somewhere in the figure? ie the slope coefficients.

Response: Thank you for your insightful suggestion. We apologize for any difficulties our initial presentation of Fig. S6 may have caused. To clarify, this figure displays the temporal evolution of the area covered by thin ice, as well as the averaged W_{ACF} and W_{PS} for the 12 subregions shown in Fig. S5. Based on your recommendation, we have added best-fitting lines that denote significant trends to the figure. Specifically, we've included the slope coefficients for ice cover (m_{ice}), W_{ACF} (m_{acf}), and W_{PS} (m_{ps}) to make the data easier to interpret. These amendments are now reflected in the revised manuscript.

L249-251: I think the mention of wavier jet stream is not necessary here. This is a hotly debated topic which I personally do not think is very strong, and I do not think the mechanisms suggested by the authors rely on it. Therefore I think including this discussion may weaken the argument.

Response: Thank you for your thoughtful feedback. We recognize the contentious nature of the discussion around a 'wavier jet stream' and agree that its inclusion in this context could potentially weaken our argument. Accordingly, we have removed the relevant paragraph from the revised manuscript to focus on the core findings and implications of our study.

L254-256: Overstated.

On a side note, I think the authors need to understand that not all changes in the Arctic are driven by climate change, and some can be attributed to internal climate variability. I think the authors should discuss this and what this means for their results.

Dörr, J. S., Bonan, D. B., Årthun, M., Svendsen, L., and Wills, R. C. J.: Forced and internal components of observed Arctic sea-ice changes, The Cryosphere Discuss. [preprint], <https://doi.org/10.5194/tc-2023-29>, in review, 2023.
England, M., A. Jahn, and L. Polvani (2019), Non-uniform contribution of internal variability to recent Arctic sea ice loss, Journal of Climate, 32, 4039-4053, doi: 10.1175/JCLI-D-18-0864.1

Response: Thank you for your thoughtful feedback. We think the reviewer means the L269-271 regarding the climate change's role in Arctic variability: “the variability of Arctic sea ice on weather time scales has significantly increased due to the melting of thick perennial sea ice. This finding underscores the profound impact of climate change on the Arctic region.”

We totally agree that not all changes in the Arctic are driven by climate change, and some can be attributed to internal climate variability. In line with your suggestion, we have revised the text as, “The variability of Arctic sea ice on weather time scales has increased, partially attributable to the melting of thick perennial sea ice. This finding underscores both the profound impact of climate change and the role of internal climate variability on the Arctic region [1,2].”

- 1. Dörr, J. S., Bonan, D. B., Årthun, M., Svendsen, L., and Wills, R. C. J.: Forced and internal components of observed Arctic sea-ice changes, The Cryosphere Discuss. [preprint], <https://doi.org/10.5194/tc-2023-29>, in review, 2023.*
- 2. England, M., A. Jahn, and L. Polvani (2019), Non-uniform contribution of internal variability to recent Arctic sea ice loss, Journal of Climate, 32, 4039-4053, doi: 10.1175/JCLI-D-18-0864.1*